# From In Silico Simulation between TGF-*β* Receptors and Quercetin to Clinical Insight of a Medical Device Containing *Allium cepa*: Its Efficacy and Tolerability on Post-Surgical Scars

**DOI:** 10.3390/life13081781

**Published:** 2023-08-21

**Authors:** Terenzio Cosio, Gaetana Costanza, Filadelfo Coniglione, Alice Romeo, Federico Iacovelli, Laura Diluvio, Emi Dika, Ruslana Gaeta Shumak, Piero Rossi, Luca Bianchi, Mattia Falconi, Elena Campione

**Affiliations:** 1Post Graduate School of Microbiology, Immunology, Infectious Diseases, and Transplants (MIMIT), Microbiology Section, Department of Experimental Medicine, University of Rome Tor Vergata, 00133 Rome, Italy; terenziocosio@gmail.com; 2Dermatologic Unit, Department of Systems Medicine, University of Rome Tor Vergata, 00133 Rome, Italy; costanza@med.uniroma.it (G.C.); lauradiluvio@yahoo.it (L.D.); ruslanagaetashumak@gmail.com (R.G.S.); luca.bianchi@uniroma2.it (L.B.); 3Virology Unit, Tor Vergata Hospital, 00133 Rome, Italy; 4Department of Surgical Sciences, University Nostra Signora del Buon Consiglio, 1000 Tirana, Albania; filadelfo.coniglione@uniroma2.it; 5Department of Biology, University of Rome Tor Vergata, Via della Ricerca Scientifica, 00133 Rome, Italy; alice.romeo@uniroma2.it (A.R.); federico.iacovelli@uniroma2.it (F.I.); falconi@uniroma2.it (M.F.); 6Dermatology, Department of Medical and Surgical Sciences Alma Mater Studiorum, University of Bologna, 40138 Bolog, Italy; emi.dika3@unibo.it; 7Oncologic Dermatology Unit, IRCCS Azienda Ospedaliero-Universitaria di Bologna, 40138 Bologna, Italy; 8Department of Surgical Sciences, University of Rome Tor Vergata, 00133 Rome, Italy; piero.rossi@uniroma2.it; 9Minimally Invasive Unit, Tor Vergata Hospital, 00133 Rome, Italy

**Keywords:** *Allium cepa*, pullulan, hyaluronic acid, scar, silicone gel, medical device, molecular docking, quercetin

## Abstract

(1) Objective: Keloid and hypertrophic scars are a challenge in clinical management, causing functional and psychological discomfort. These pathological scars are caused by a proliferation of dermal tissue following skin injury. The TGF-β/Smad signal pathway in the fibroblasts and myofibroblasts is involved in the scarring process of skin fibrosis. Today, multiple therapeutic strategies that target the TGF-β/Smad signal pathway are evaluated to attenuate aberrant skin scars that are sometimes difficult to manage. We performed a head-to-head, randomized controlled trial evaluating the appearance of the post-surgical scars of 64 subjects after two times daily topical application to compare the effect of a class I pullulan-based medical device containing *Allium cepa* extract 5% and hyaluronic acid 5% gel versus a class I medical device silicone gel on new post-surgical wounds. (2) Methods: Objective scar assessment using the Vancouver Scar Scale (VSS), POSAS, and other scales were performed after 4, 8, and 12 weeks of treatment and statistical analyses were performed. The trial was registered in clinicalTrials.gov ( NCT05412745). In parallel, molecular docking simulations have been performed to investigate the role of *Allium cepa* in TGF-β/Smad signal pathway. (3) Results: We showed that VSS, POSAS scale, itching, and redness reduced significantly at week 4 and 8 in the subjects using devices containing *Allium cepa* and HA. No statistically significant differences in evaluated scores were noted at 12 weeks of treatment. Safety was also evaluated by gathering adverse events related to the application of the gel. Subject compliance and safety with the assigned gel were similar between the two study groups. Molecular docking simulations have shown how *Allium cepa* could inhibit fibroblasts proliferation and contraction via TGF-β/Smad signal pathway. (4) Conclusions: The topical application of a pullulan-based medical device containing *Allium cepa* and HA showed a clear reduction in the local inflammation, which might lead to a reduced probability of developing hypertrophic scars or keloids.

## 1. Introduction

In recent years, dermo-cosmetics has been rapidly spreading into the market for the treatment of post-surgical scars—namely for hypertrophic and keloid scars—as these have a high socio-cultural impact. After surgery, scars are problematic for both surgeons and patients. At first, surgical scars can also cause pain, itching, discomfort, contracture, and other functional impairment [1,2]. Moreover, surgical wound healing without noticeable scarring is an important aspect of cosmetics, significantly affecting the patients’ quality of life. Topical applications in this field are increasingly pushing on the use of biocompatible molecules in order to decrease the effects related to the use of mineral oil derivatives that can lead to skin atrophy or to the development of granulomatous reactions [3]. Hypertrophic scars represent a condition of dysregulation in the tissue repair process with an excessive proliferation of fibroblasts and terminal differentiation into myofibroblasts. In recent years, the understanding of the pathogenic mechanisms that drive keloid and hypertrophic scar development and progression has grown rapidly. These pathological scars are essentially the result of chronic inflammations of the injured reticular dermis. Besides inflammation, mechano-signaling pathways play an important role in scar formation [4,5]. Therefore, treatment strategies against these scars should focus on preventing or dampening inflammation. Since hypertrophic scars are not tumors, the therapeutic target should be the blood vessels, the endothelial cells, or the perivascular cells, rather than the fibroblasts [6]. On the other hand, keloids, classified as benign neoplastic lesions, with a higher percentage in the African American (AA) population, tend to relapse, thus resulting in difficult treatments and discomfort for patients. Keloids and hypertrophic scars develop due to the hyperproliferation of dermal fibroblasts, although only keloids are characterized by an increase in the rate of proliferation of fibroblasts [1]. The most well-known mechanism that regulates collagen synthesis in fibroblasts and myofibroblasts is the TGF-β/Smad signaling pathway [7]. The increased collagen production in aberrant scars is due to the long-term overactivation of fibroblasts and myofibroblasts brought on by the persistent activation of the TGF-β/Smad signaling pathway. Two types of treatment approaches are used to target the TGF-β/Smad signaling pathway in fibroblasts and myofibroblasts in order to try to suppress cell proliferation and interfere with cellular activities [7]. The treatment of keloids and hypertrophic scars is widely mentioned in the literature and several techniques and methods have been used. Many treatment modalities can be used and, interestingly, they all act by reducing inflammation and blocking TGF-β/Smad signaling. They include corticosteroid injection/tape/ointment, radiotherapy, cryotherapy, compression therapy, stabilization therapy, 5-fluorouracil (5-FU) therapy, and surgical methods that reduce skin tension. Topical non-surgical treatment silicone gel [8], 5-fluorouracil [9], onion extract gel [10,11], imiquimod, verapamil, mitomycin C, green tea extracts, *Aloe vera* and vitamin E [12] are used in clinical practice and are reported in the literature. The treatments should work at a biologic level and try to produce a cosmetic and complication-free management strategy. In this view, quercetin, the main flavonoid compound contained in *Allium cepa*, has been reported to play a crucial role in modulating inflammation and signaling responses promoting tissues fibrosis [13]. To date, the aim of our study is to compare the effect of a topical class I pullulan-based medical device containing *Allium cepa* and HA gel versus a class I medical device silicone gel on new post-surgical wounds to prevent hypertrophic scar formation. Moreover, we performed molecular docking simulations between quercetin and the TGF-β Receptor type 1 (TbrI) and type 2 (TbrII) to evaluate a possible direct effect of *Allium cepa* extract on TGF-β/Smad signaling. 

## 2. Materials and Methods

### 2.1. Study Design and Enrolled Population

We performed a head-to-head, randomized controlled trial evaluating the appearance of post-surgical scars in 64 subjects for 12 weeks in order to compare the effect of a class I pullulan-based medical device containing *Allium cepa* 5% and HA 5% gel versus a class I medical device silicone gel on new post-surgical wounds. The 64 subjects were distributed randomly in the two groups: 30 applied topical class I medical device silicone gel and 34 applied the class I pullulan-based medical device containing *Allium cepa* and HA gel for three months. The sample size was calculated using two independent means, which were acquired from previous research [14,15]. The randomization sequence was created using Excel version 2306, with a 1:1 allocation of the subject in one of the two arms by sub-investigator G.C. Sub-investigator T.C. enrolled the participants in the dermatological clinic of the Tor Vergata University Hospital. Sub-investigator R.G. assigned participants to interventions using simple randomization in Excel version 2306. Medical device samples, previously numbered by sub-investigator G.C., were given to subjects by sub-investigator T.C. The application of the products was recorded by sub-investigator T.C. and R.G by means of a paper diary provided to the subjects in which they reported the application and/or non-application of the medical devices. Furthermore, during the enrolment visit, sub-investigator T.C. explained how to apply the medical device (after cleansing the area, a layer of medical device is applied with circular movements until completely absorbed). The study was developed in 10 months, including recruitment, evaluation every 4 weeks for 12 weeks, and an evaluation of the secondary objective in the third month (T3). Data acquisition and analysis were performed in compliance with protocols approved by the Ethical Committee of the Tor Vergata University Hospital, Rome (ethical approval number 211.19). Written informed consent was obtained from all participants prior to the study. The study was conducted in accordance with the ICH/GCP guidelines and in compliance with the current regulations relating to studies on devices. The study complied with the tenets of the Declaration of Helsinki. The trial was registered in clinicalTrials.gov (https:clinicaltrials.gov/ct2/show/NCT05412745 accessed on 14 September 2022).

### 2.2. Inclusion and Exclusion Criteria

Inclusion criteria: subjects aged between 18 and 70 who underwent surgery for the excision of skin lesions at least 5 days or 20 days before the start of the protocol, respectively, for surgical procedure on face or limbs and trunk. Exclusion criteria: Subjects affected by spontaneous keloids; diabetic subjects with a previous history of disorders in the repair of wounds; subjects with over-infected wounds after the first week from surgery; subjects with documented sensitivity to silicone gel; and subjects affected by collagen disorders (e.g., pseudoxanthoma elasticum, poikilodermatosis). Moreover, the criteria for the withdrawal of subjects from the trial included: failure to adhere to topical therapy; replacement of the trial topical treatment; and voluntary decision by the subject. 

### 2.3. Endpoints and Parameters Evaluated

Since the medical device containing *Allium cepa* presents a high concentration of quercetin, our primary endpoint was to achieve the confirmation of the efficient interaction between quercetin and TGF-beta receptors and a parallel clinical efficacy. The primary endpoint was the evaluation of the effectiveness of the class I medical device containing onion (*Allium cepa*) extract gel compared to the silicone gel used in the topical treatment of the post-surgical wound for the prevention of hypertrophic scars. Objective scar assessment using the Vancouver Scar Scale (VSS), Manchester Scar Scale, Patient and Observer Scar Assessment Scale (POSAS), itching, redness, and pliability were performed after 4 (T1), 8 (T2), and 12 (T3) weeks of treatment (Appendix A). Secondary endpoint evaluated the tolerability at 4, 8, and 12 weeks after application. Local tolerability was measured by the investigator on a 5-point scale, where 1 = excellent (no functional or physical sign from examination), 2 = very good (transitory functional signs and no physical signs from examination), 3 = good (transitory or persistent functional signs with transitory physical signs), 4 = moderate (persisting functional or physical signs that require modified administration but not discontinuation), and 5 = bad (functional or physical signs that require discontinuation). 

### 2.4. Molecular Docking Simulations

Crystal structures of the kinase domains of the TGF-β Receptor type 1 (TbrI) and type 2 (TbrII) were obtained from the PDB database (PDBIDs: 3TZM and 7DV6) [14,15]. Ligands crystallized with the proteins were removed before the docking procedure. Quercetin structure was retrieved from the PubChem compound database (PubChem CID: 5280343) [16]. Molecular docking simulations were performed using the AutoDock Vina 1.2 program [17], and the prepare_receptor4.py and prepare_ligand4.py tools of the AutoDockTools4 program [18] were used to convert the receptors and drug structure files into pdbqt format. For both receptors, the simulation box was centered over their ATP-binding pocket, having a size of x = 26.3 Å, y = 23.6 Å, z = 27.4 Å for the TbrI and of x = 24.0 Å, y = 25.9 Å, z = 23.6 Å for the TbrII. To improve the accuracy of binding prediction, 17 receptor residue side chains surrounding the binding site have been considered as flexible for the TbrI (Ile211, Lys213, Arg215, Val219, Lys232, Glu245, Tyr249, Leu260, Leu278, Ser280, Tyr282, His283, Lys335, Lys337, Asn338, Leu340, Asp351) and 13 for the TbrII (Val250, Val258, Lys277, Leu305, Thr325, Phe327, His328, Asn332, Ser383, Asn384, Leu386, Cys396, Asp397). Three molecular docking simulations, each including ten docking runs, were then performed for each receptor, and final interaction energies were calculated as an average over the three replicas. Redocking simulations of the crystallized inhibitors were performed using a box of size x = 15.0 Å, y = 13.5 Å, z = 18.4 Å for TbrI and x = 16.9 Å, y = 17.3 Å, z = 18.8 Å for TbrII, centered around the ligands’ binding sites. Interaction analyses on the best binding poses obtained for quercetin were performed using the Ligand Interaction tool of the Schrödinger Maestro software (Schrödinger Release 2023-1, 2023).

### 2.5. Statistical Analysis

The statistical analysis of the data was carried out by applying parametric or non-parametric tests depending on the distribution of the data that was going to be obtained. Results were reported as average ± SE, median values or percentages, considering parameter type. Non-parametric Krustal–Wallis test was used to evaluate the variation of clinical parameters in the pre-established observation times (T0, T1, T2, and T3) for each treatment, whereas comparison between treatments was performed via Mann–Whitney analysis and chi-squared test (χ^2^ test). The study was conducted on a group of subjects selected through specific inclusion and exclusion criteria according to the scientific rationale of the protocol. SPSS v.27 software was used for statistical analyses. The differences were considered statistically significant for *p*-values < 0.05.

## 3. Results

### 3.1. Characteristics of Enrolled Subjects 

A total of 64 subjects were enrolled in the study. They were randomized in the two groups: the first one (group A) included topical treatment with silicone gel, the second one (group B) used a topical medical device with a pullulan-based medical device containing *Allium cepa* and HA gel (Figure 1). 

In group A, 30 subjects were enrolled, with a mean age of 50 ± 7.7 years, including 14 males and 16 females. Twenty-eight subjects presented one scar, while two presented two or more scars. The most frequent site of intervention was the trunk (43.3%). In Group B, there were 34 subjects, with a mean age of 51.5 ± 7.7 years, including 14 males and 20 females. Thirty subjects presented one scar, while four presented two or more scars (Table 1). The two groups were homogeneous per age, sex, number of post-surgical scars and diet modification (Table 1; statistical analysis *p* < NS). Moreover, all subjects followed the protocol thanks to the simplicity of the application of the products in the studio and the post-surgery aesthetic interest.

Even in this case, the most frequent side of intervention was the trunk (Figure 2 and Figure 3). No significant differences between groups A and B’s demographical features were noticed. 

### 3.2. Comparison of Clinimetric Scores

The analysis of the population data (Figure 4) showed that the observed POSAS was already statistically significant after T1 and remained at all time points for groups A and B’s treatments (ANOVA, *p* < 0.01). Differences were statistically significant between treatments at T1 and T2 (* *t*-test; *p* < 0.05). Treatment with class I pullulan-based medical device containing *Allium cepa* and HA gel was shown to be statistically superior in reducing the POSAS patient score at all time points (gray bars, ANOVA; *p* < 0.01). Moreover, comparison between the two treatments showed a significant difference, at T1 and T2 (* *t*-test; *p* < 0.05), as reported in Figure 3. Treatment with class I pullulan-based medical device containing *Allium cepa* and HA was not only statistically superior in reducing redness (post-operative redness and inflammation, gray bars, ANOVA *p* < 0.01), but also resulted in a reduction in the two parameters after treatment for both groups A and B (gray bars, ANOVA *p* < 0.05). At T3, there were no statistically significant differences between the two groups (* *t*-test *p* < 0.01) (Figure 4). Treatment with onion extract gel also showed a better reduction on the Vancouver Scale and itching (gray bars, ANOVA *p* < 0.05). At T1 and T2, there were statistically significant differences between the two groups (* *t*-test *p* < 0.05) (Figure 4). Treatment with class I pullulan-based medical device containing *Allium cepa* and HA was effective in post-surgical lesions regardless of the body site. This highlights how its effectiveness is maintained in all body sites, and thus makes it suitable for any type of scar, regardless of the site of the operation. In our head-to-head, randomized study, POSAS observed, POSAS patients, Vancouver Score, itching, and redness were statistically different in group B compared to A (Figure 4 and Figure 5). Local tolerability, measured by the investigator on a 5-point scale, did not show any statistical difference between groups A and B at all timepoints analyzed (Figure 6; NS). Moreover, no subject at any follow-up visit had any adverse events or local reactions. 

### 3.3. Molecular Docking Simulations of TbrI and TbrII Kinase Domains in Complex with Quercetin

Molecular docking simulations have been performed to evaluate the interaction of quercetin, the main flavonoid compound contained in *Allium cepa*, with the kinase domain of the human TGF-β Receptor type 1 (TbrI) and type 2 (TbrII) (PDBIDs: 3TZM and 7DV6) [16,17]. In fact, these receptors are involved in a variety of pathological processes, including the signaling responses promoting tissues fibrosis [13]. A specific pocket at the kinases’ ATP-binding domains was evaluated as a putative quercetin binding site, being the target of other small-molecule inhibitors known to interfere with TGFβ signaling [17,18,19,20,21]. Three molecular docking simulations were performed for each receptor, showing that quercetin has an average interaction energy of −10.7 ± 0.2 kcal/mol for TbrI and −9.6 ± 0.4 kcal/mol for TbrII. The best binding poses obtained from the docking simulations and the main interactions established by the compound within the two pockets are shown in Figure 6 and Figure 7. Quercetin achieved the same binding location in the two pockets, although in the TbrI–quercetin complex, the catechol moiety of the compound was oriented toward the interior of the pocket, while an opposite orientation was observed for the compound within the TbrII active site (Figure 7). Quercetin established a total of 19 interactions with both TbrI and TbrII, including one hydrogen bond with a His residue of both receptors (Figure 8). Remarkably, the presence of a hydrogen bond with the TbrI His283 residue is a feature common to many ATP-competitive kinase inhibitors [17]. 

Despite the low sequence identity of the two receptors, the ATP-binding site and the adjacent regions were more conserved, showing either identical residues or preserved interaction types. Consequently, the main interactions established by quercetin were similar in the two pockets and mostly involved structurally superimposable residues (Figure 9). This suggests that this compound could equally target the two receptors, although showing a higher preference for the TbrI active pocket as indicated by the interaction energies calculated by the docking simulations. 

Notably, the best binding poses obtained closely resemble those achieved by the two kinase inhibitors crystallized within the proteins’ active sites, namely SB431542 (PDBID: 3TZM) [17] and 5-[(3S)-5,5-dimethyloxolan-3-yl]-6-methoxy-3-(2-methoxypyridin-4-yl)pyrazolo [1,5-a]pyrimidine (PDBID: 7VSB) [18]. A superposition of the complexes highlights similar ligands orientations, contacted residues, and interaction types between the crystallized and docked structures. The redocking of these inhibitors within the binding pockets resulted in interaction energies similar to those calculated for quercetin, evaluating −11.4 and −9.3 kcal/mol for the TbrI–inhibitor and TbrII–inhibitor complexes, respectively.

## 4. Discussion

Scar prevention in the early stages of wound healing is an essential aspect of care. Ongoing scar management is generally conducted through self-care, using non-invasive methods such as silicone. Although silicone is the first-line non-invasive, prophylactic, and therapeutic measure for scar management [22,23,24], the issues related to the use of these products, as well as patient research for more effective alternatives, favored the raising of new products made to improve outcomes. Amongst polysaccharides studied for skin regeneration, pullulan is the major exopolysaccharide synthesized intracellularly by the polymorphic fungus *Aureobasidium pullulans* and further secreted out to the cell surface, consisting of three glucose units connected by α−1,4 glycosidic bonds (maltotriose) and consecutive maltotriose units connected by α−1,6 glycosidic bonds [25]. Pullulan has attracted attention for its anti-inflammatory, anti-bacterial, and antioxidant properties, as well as its lack of immunogenicity and low cost [26]. Given its physical–chemical characteristics, it leads to the formation of a barrier. It is non-toxic, non-carcinogenic, non-mutagenic, non-immunogenic [27,28,29], and odorless, and is therefore also suitable for human use via topical formulation in cases of sensitivity to other active ingredients or in cases of chemical polysensitivity. Pullulan is biodegradable [30] and is used as a carrier associated with HA to prevent the denaturation of protein molecules [31] and therefore guarantees a better function of the *Allium cepa* [32]. Creating novel cosmetic solutions that can penetrate this barrier is essential due to effective active ingredients and efficient carriers. The chosen ingredients must be released at the appropriate skin layers to restore intercellular communication, maintain stratum corneum hydration, and renew the native ECM structure of the dermis when this becomes diseased or sagged with age [33]. This activity requires the use of proper active components and carriers, such as pullulan and its composites, which may regenerate the structural architecture of damaged or ageing skin when changed or destroyed by various illnesses or injuries. Li et al. [34] created pullulan polymers with grafted hyaluronic acid and used them to create innovative biocompatible wound healing films. Vora et al. [35] investigated pullulan-based dissolving microneedle arrays for an improved transdermal delivery of small and large macromolecules. In our study, the product under examination based on *Allium cepa* and HA, also including pullulan with silicone-like effect, demonstrated its non-inferiority compared to silicone. Moreover, the product has no environmental and biodegradable impact. Onion, *Allium cepa* L. (*Alliaceae*), contains many organosulfureous compounds, flavonoids (mainly quercetin in the form of aglycone or sugar conjugates), saponins, and vitamins (B1, B2, B5, E). Onion extract, mainly composed of quercetin, is believed to reduce the level of histamine, inflammation, and collagen production in abnormal scars [36]. The use of plant extracts in wound healing is associated with a satisfying outcome in most cases. In this field, flavonoids are generally abundant in natural extracts and the high content of quercetin is an interesting feature. In fact, quercetin has been studied in various wound healing conditions, alone or combined with other phytochemicals, proving to be a valid option in the management of wound repair [37]. *Allium cepa* presents different active molecules, including gallic acid, quercetin, protocatechuic acid, and kaempferol [38]. Among them, quercetin is the major flavonoid component with concentrations ranking until 5110 µg/g, more than 40-folds the concentration of other isolated compounds such as gallic acid and kaempferol. Moreover, previous publications investigated the role of quercetin in wound healing and scar prevention. Karakaya et al. [39] evaluated the ethnopharmacological use of *Epilobium angustifolium*, *E. stevenii*, and *E. hirsutum* by using in vivo and in vitro experimental models, as well as to identify the active wound-healer compound(s) and to explain the probable mechanism of the wound-healing activity. Their in vitro tests showed that quercetin and its glucosides could be the compound responsible for the wound-healing activity due to their significant anti-hyaluronidase, anti-collagenase, and antioxidant activities. In parallel, Özbilgin et al. [40] evaluated *E. characias* subsp. *wulfenii* extract in wound healing. Quercetin and its glucoside derivates were found to be the highest compound in the extract of *E. characias* subsp. *wulfenii*, which might be the active principles in anti-inflammatory and wound-healing activities. Mi et al. [41] evaluated the effect of quercetin on wound healing by analyzing wound healing rate of pro-inflammatory cytokines in mice. Their observations come from the use of *Oxytropis falcata*, a wild-growing plant mainly distributed in an altitude of 2700–4300 m in China, used in treating inflammation, injury, and bleeding. They demonstrated that quercetin could promote the proliferation and migration of fibroblasts and HaCat cells. Furthermore, quercetin has been shown to promote wound healing in mice by inhibiting the inflammatory response and increasing the expression of the growth factor, as well as Wnt/β-catenin pathway and telomerase RNA catalytic elements (TERT). Onion extracts can inhibit the production of IL-6 and vascular endothelial growth factor (VEGF) in the cell line of skin fibroblasts [42]. These cytokines could play a fundamental role in the abnormal growth of fibroblasts in keloids [43]. Human dermal fibroblasts use b1 integrins primarily for adhesion to fibronectin and collagen fibers [44]. During scar formation, dermal fibroblasts transiently expressed mainly a2 and a5 integrin subunits, which in turn are associated with the upregulation of the b1 subunit. The upregulation of some chains of integrins is also recognized as an important regulator of the deposition of the extracellular matrix (ECM) components associated with wound healing [45]. It has also been shown that integrin b1 expressed by fibroblasts is necessary for fibrogenesis. Furthermore, it is interesting to note that the downregulation of B1 integrin caused resistance to skin scleroderma in the mouse model [46]. In addition, the onion extract induced apoptosis at 1000 mg/mL in fibroblast cultures and, depending on the concentration, downregulated the expression of integrin b1 on human fibroblasts. Apoptosis has played an important role in the proper healing of wounds and can signal the end of an inflammatory phase of this process [32]. Apoptosis-related genes have been reported to be downregulated in the cells that make up keloids [47]. Given the high concentration of flavonoids in *Allium cepa* extract, it has been reported that quercetin—a flavonoid—inhibits the proliferation and contraction of fibroblasts isolated from keloids [48] in in vitro studies. This seems to block TGF-β—that inhibits the expression of the receptor and the nuclear translocation of SMAD2/3—which in turn alters the expression of collagen [49] as well as of the IGF-1 signal, through the reduction in the receptor and intracellular signaling, thus influencing the proliferation of keloid fibroblasts and reduces the contraction of collagen [50]. In addition, onion extract upregulates matrix metalloproteinase-1 (MMP-1) [51]—as seen in vitro and in vivo studies of hairless mice—on human skin fibroblasts with topical administration of the ointment. However, it is statistically ineffective in improving scar height and itching [50].

To confirm a possible direct effect of quercetin on TGF-β/Smad signaling, we performed molecular docking simulations between quercetin and the TGF-β Receptor type 1 (TbrI) and type 2 (TbrII). Our results suggest that quercetin could efficiently bind the ATP-binding domain of both TbrI and TbrII and consequently interfere with the signaling cascade mediated by the two receptors. Thus, these results provide another possible molecular mechanism explaining the quercetin inhibition of fibroblasts proliferation and contraction, previously observed in in vitro studies [48,52], and sustain the favorable results obtained by using medical devices containing *Allium cepa* and HA to prevent hypertrophic scar formation in post-surgical wounds. From all these data, onion extract therapy could be used when combined with an occlusive silicone-like dressing, such as pullulan, to achieve a satisfactory decrease in scar height. In our study, the class I pullulan-based medical device reduced wound time compared to silicone, both accelerating the reduction in itching, redness, cosmetic aspect of the scar, and discomfort in treated subjects. No local adverse events were reported.

## 5. Study Limitation

Study limitations included the small sample size, the absence of a blinded evaluation, and the lack of a control group treated with a class I pullulan-based medical device and HA gel without *Allium cepa*. Regarding the study population, no other ethnicity than Europeans were included. Moreover, only four body sites were evaluated for 12 weeks with no documented photos in other follow-up visits. Moreover, the scar maturation evaluated at 12 weeks was partial and could not reflect the long-term results, and after 12 weeks, the two evaluated treatments seemed to be comparable in terms of VSS, Manchester Scar Scale, POSAS, itching, redness, and pliability. One of the major limitations of this study is also the evaluation of the *Allium cepa* extract and not the quercetin as a single molecule. Further studies are needed to evaluate the long-term clinical response and quercetin alone in real-word settings.

## 6. Conclusions

Molecular docking simulations have shown how *Allium cepa* could inhibit fibroblasts proliferation and contraction via TGF-β/Smad signal pathway. Moreover, our study demonstrated the superiority of a class I pullulan-based medical device containing *Allium cepa* and HA extract compared to a class I medical device silicone gel in the treatment of post-surgical scars with an improvement of the Vancouver Scale, POSAS patient, and observed redness and itching in a period of three months after surgery. Innovative, biobased, and biodegradable tissues and films, as well as the soft and hard packaging for their specific structures and the natural ingredients selected, may slow down, partially or globally, the skin aging phenomena and reduce or eliminate the great waste invading the planet.

## Figures and Tables

**Figure 1 life-13-01781-f001:**
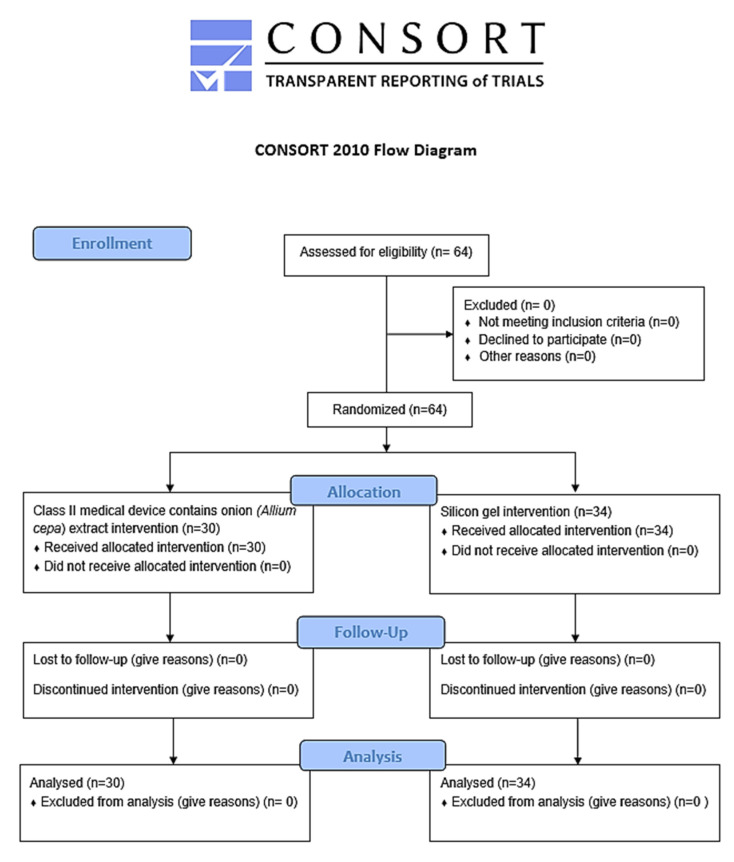
CONSORT flow chart.

**Figure 2 life-13-01781-f002:**
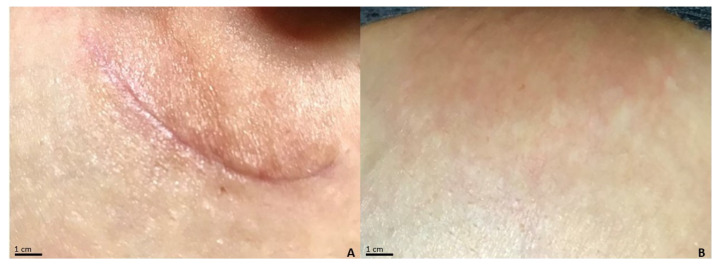
Postoperative lesion at the left periareolar level of a subject treated with a class I pullulan-based medical device containing *Allium cepa* and HA gel. (**A**) The lesion has a POSAS observer score of 13/60 at baseline; POSAS patient 19/60; Vancouver Scale 2/18; Manchester Scale 7/12. (**B**) After treatment for 3 months, the lesion presents POSAS observer 6/60; POSAS patient 7/60; Vancouver Scale 0/18; Manchester Scale 5/12.

**Figure 3 life-13-01781-f003:**
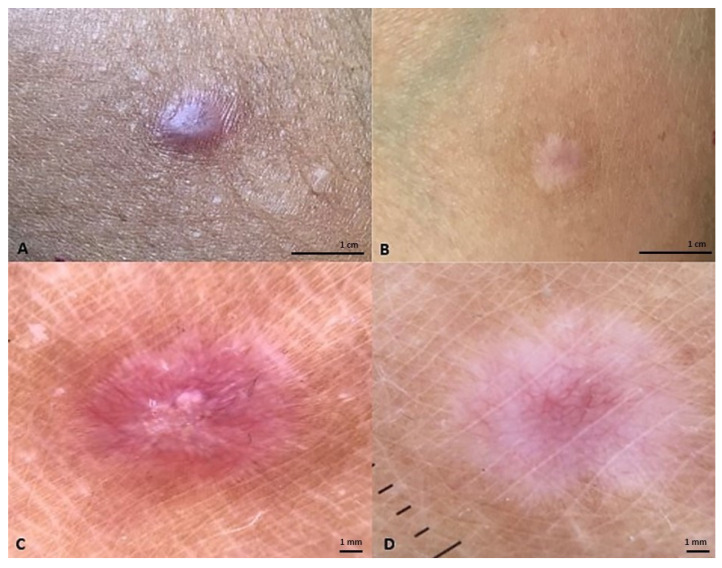
(**A**) Postoperative lesion at the upper limb level of a subject treated with a class I pullulan-based medical device containing *Allium cepa* and HA gel. The lesion has a POSAS observer score of 11/60 at baseline; POSAS patient 10/60; Vancouver Scale 6/18; Manchester Scale 1/12. (**B**) After treatment for 3 months, the lesion presents POSAS observer 10/60; POSAS patient 9/60; Vancouver Scale 6/18; Manchester Scale 1/12. (**C**) Dermoscopy of the lesion after surgery. (**D**) Dermoscopy of the lesion after 3 months of topical onion extract gel.

**Figure 4 life-13-01781-f004:**
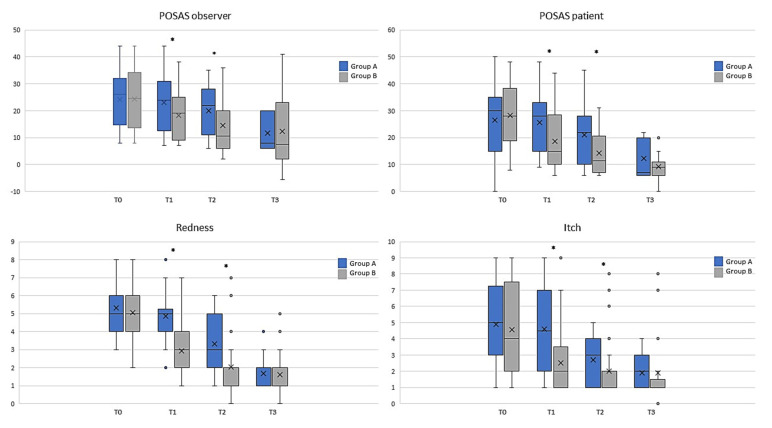
Graph bars showed that several parameters were reduced, especially in T1 and T2 between groups A (black; silicone gel) and B (gray; *Allium cepa* and HA gel). Significant differences are highlighted by Krustal–Wallis and Mann–Whitney tests. * *p* < 0.05. X, mean value.

**Figure 5 life-13-01781-f005:**
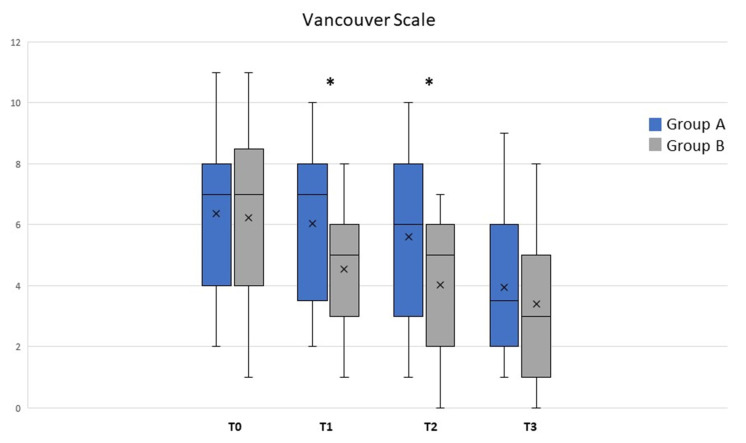
Vancouver Scale was analyzed between groups A (black; silicone gel) and B (gray; *Allium cepa* and HA gel) at different timepoints. Differences statistically significant are reported in text. * *p* < 0.05; X, mean value.

**Figure 6 life-13-01781-f006:**
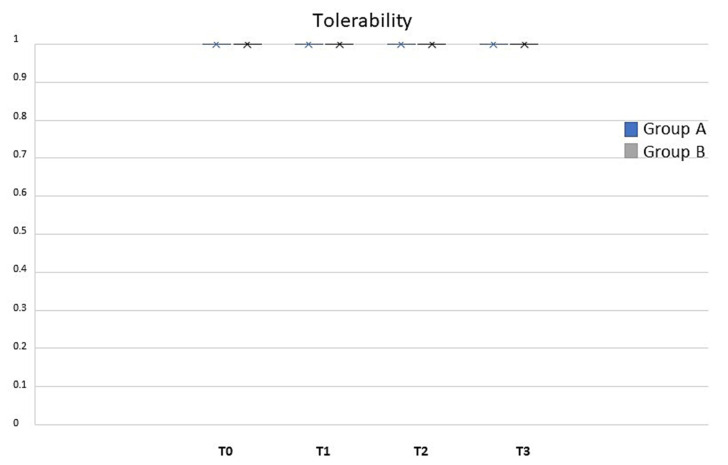
Tolerability between groups A (black; silicone gel) and B (gray; *Allium cepa* and HA gel) at different timepoints. Local tolerability presented a score of 1 at all follow-up visits both in the case and in the control group. No statistical difference has been noted (*p*-value NS). X, mean value.

**Figure 7 life-13-01781-f007:**
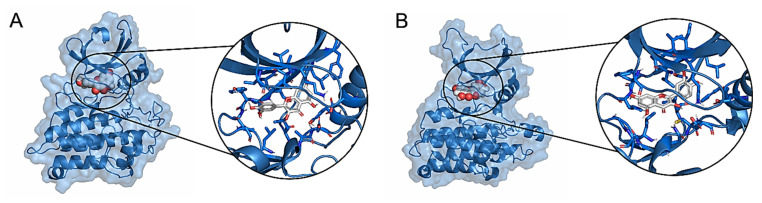
Best molecular docking complexes obtained between quercetin and the TbrI (**A**) or TbrII (**B**) kinase domains. The proteins are shown as cartoons surrounded by a transparent surface, while the ligands are represented as spheres, colored by atom type. Binding poses are magnified in the circles, where ligands and their main interacting residues are shown as sticks.

**Figure 8 life-13-01781-f008:**
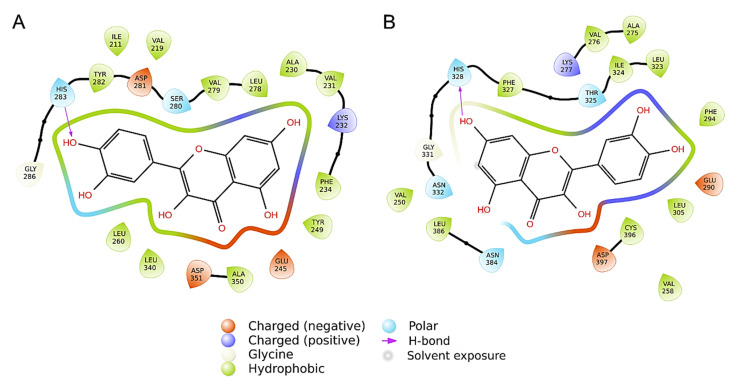
Ligand interaction patterns established within the TbrI (**A**) or TbrII (**B**) binding pockets, calculated by the Ligdiag tool of the Schrödinger Maestro software (Schrödinger Release 2023-1, 2023). Residues are colored according to the type of interaction established, as indicated by the legend.

**Figure 9 life-13-01781-f009:**
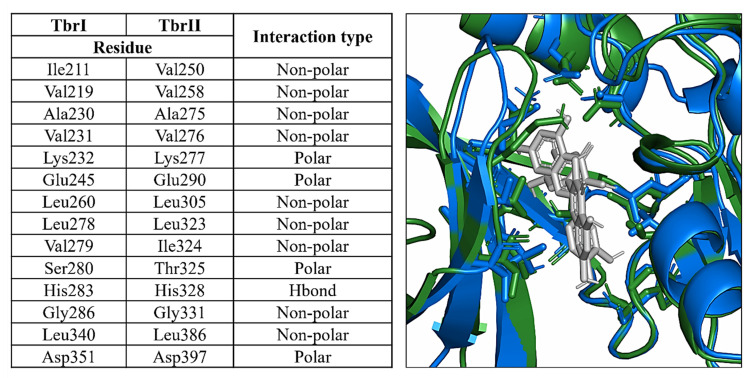
On the left, structurally overlapping residues contacted by quercetin within the TbrI and TbrII active pockets. The interaction types were derived from Figure 7. On the right, a superposition of the two protein–ligand complexes, where the ligands and the residues reported in the table are shown as sticks. TbrI and TbrII are shown as cartoons colored in blue and green, respectively.

**Table 1 life-13-01781-t001:** Demographic and clinical pathological features of the enrolled subjects in groups A (silicone gel) and B (*Allium cepa* and HA gel). The *t*-test was performed for continuous variables; χ^2^ test was performed on dichotomous variables. SE: Standard error.

	Group A	Group B	Statistical Analyses
Number	30	34	
Mean age (±SE)	50 ± 7.1	51.5 ± 7.7	*p* < 0.60
Sex			*p* < 0.65
Male	14	14	
Female	16	20	
n lesions			*p* < 0.48
1	28	30	
2	2	4	
Diet modification			*p* < 0.34
No	30	31	
Yes	0	3	
Hypertrophic scar			*p* < 0.56
No	26	31	
Yes	4	3	
Lesion site			*p* < 0.83
Trunk	13	16	*p* < 0.68
Face	4	7	
Upper limbs	3	3	
Lower limbs	7	8	

## Data Availability

Data sharing not applicable.

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
