# Peer review of "From In Silico Simulation between TGF-*β* Receptors and Quercetin to Clinical Insight of a Medical Device Containing *Allium cepa*: Its Efficacy and Tolerability on Post-Surgical Scars"

_life, 2023, doi:10.3390/life13081781_

Round 1

Reviewer 1 Report

In this article, Cosio et al performed a head-to-head, randomized, pivotal controlled trial evaluating the appearance of the post-surgical scars of 64 subjects after two times daily application to compare the effect of a class I pullulan based medical device containing Allium cepa and hyaluronic acid gel versus a class I medical device silicone gel on new post-surgical wounds. They showed that VSS, POSAS scale, itching and redness reduced significantly at week 4 and 8 in subjects using devices containing Allium cepa and HA. They also performed molecular docking simulations, which have shown how Allium cepa could inhibit fibroblasts proliferation and contraction via TGF-β/Smad signal pathway. They concluded that the topical application of a pullulan based medical device containing Allium cepa and HA showed a clear reduction in the local inflammation, which might lead to a reduction in the probability of developing hypertrophic scars or keloids. The results of the clinical trials, although small in number, are presented together with data from in silico that support the results and are considered more convincing. I have some questions.

major  concerns)

1) I think the weak point in this study is why you focused on quercetin, in Allium cepa. This is because that there are many different components in Allium cepa. It would be more convincing if you could show that among the various components, quercetin is the most effective.

minor concerns)

1) In line 28, "wounds,." means "wounds."

Author Response

First of all,

we would like to thank both reviewers for their constructive comments. Below are our answers, item by item.

With our best regards

In this article, Cosio et al performed a head-to-head, randomized, pivotal controlled trial evaluating the appearance of the post-surgical scars of 64 subjects after two times daily application to compare the effect of a class I pullulan based medical device containing Allium cepa and hyaluronic acid gel versus a class I medical device silicone gel on new post-surgical wounds. They showed that VSS, POSAS scale, itching and redness reduced significantly at week 4 and 8 in subjects using devices containing Allium cepa and HA. They also performed molecular docking simulations, which have shown how Allium cepa could inhibit fibroblasts proliferation and contraction via TGF-β/Smad signal pathway. They concluded that the topical application of a pullulan based medical device containing Allium cepa and HA showed a clear reduction in the local inflammation, which might lead to a reduction in the probability of developing hypertrophic scars or keloids. The results of the clinical trials, although small in number, are presented together with data from in silico that support the results and are considered more convincing. I have some questions.

Dear Reviewer, thank you for your interest in our manuscript and for the profitable comments to improve its quality.

  • Major concerns

I think the weak point in this study is why you focused on quercetin, in Allium cepa. This is because that there are many different components in Allium cepa. It would be more convincing if you could show that among the various components, quercetin is the most effective.

  • Dear reviewer, thank you for your suggestion. When in vivo studies are carried out on natural extracts and not on single molecules, one of the problems that arises is always demonstrating the effect of a single compound with respect to the whole composition. In our case, in order to strengthen the hypothesis that quercetin is the molecule involved in the prevention of the onset of post-surgery hypertrophic and keloid scars, we improved and deepened this aspect first of all by focusing on the molecules presented in Allium cepa We then evaluated how quercetin, derived from other natural sources, was used in wound healing and to prevent scars, to force the hypothesis that quercetin is not only the major flavonoid compound presented in the Allium cepa extract, but that it is also studied in the same fields but from different sources.

The use of plant extracts in wound healing is associated with a satisfying outcome in most cases. In this field, flavonoids are generally abundant in natural extracts and the high content of quercetin is an interesting feature. In fact, quercetin has been studied in various wound healing conditions, alone or combined with other phytochemicals, proving to be a valid option in the management of wound repair [37]. Allium cepa presents different active molecules including gallic acid, quercetin, protocatechuic acid and kaempferol [38]. Among them, quercetin, is the major flavonoid component with concentrations ranking until 5110 µg/g, more than 40 folds the concentration of other isolated compounds such as gallic acid and kaempferol. Moreover, previous publications investigated the role of quercetin in wound healing and scar prevention. Karakaya et al. [39] evaluated the ethnopharmacological use of Epilobium angustifolium, E. stevenii and E. hirsutum by using in vivo and in vitro experimental models, and to identify the active wound-healer compound (s) and to explain the probable mechanism of the wound-healing activity. Their in vitro tests showed that quercetin and its glucosides could be the compound responsible for the wound-healing activity due to their significant anti-hyaluronidase, anti-collagenase, and antioxidant activities. In parallel, Özbilgin et al. [40] evaluated E. characias subsp. wulfenii extract in wound healing. Quercetin et its glucoside derivates were found to be the highest compound in the extract of E. characias subsp. wulfenii, which might be the active principles in anti-inflammatory and wound-healing activities. Mi et al. [41] evaluated the effect of quercetin on wound healing by analyzing wound healing rate, pro-inflammatory cytokines in mice. Their observations come from the use of Oxytropis falcata, a wild-growing plant mainly distributed in an altitude of 2700–4300 m of China, used in treating inflammation, injury and bleeding. They demonstrated that quercetin can promote the proliferation and migration of fibroblasts and HaCat cellsl. Furthermore, quercetin has been shown to promote wound healing in mice by inhibiting the inflammatory response and increasing the expression of the growth factor, as well as Wnt/β-catenin pathway and telomerase RNA catalytic elements (TERT).

Moreover, we added your suggestion in the study limitation section.

Study limitation included a small sample size, no other ethnicity than European, only four body sites evaluation of the lesions at baseline and after 12 weeks with no documented photos in other follow-up visits. Moreover, the scar maturation evaluated at 12 weeks is partial and could not reflect the long-term results. Moreover, after 12 weeks the two evaluated treatments seem to be comparable in terms of VSS, Manchester Scar Scale, POSAS, itching, redness and pliability. One of the major limitations of this study is also the evaluation of the Allium cepa extract and not of quercetin as single molecule. Further studies are needed to evaluate the long-term clinical response of quercetin alone in real-word settings.

  • Minor concerns

In line 28, "wounds,." means "wounds."

Dear reviewer, thank you for your suggestion. The manuscript has been corrected.

Reviewer 2 Report

The authors present and interesting paper reporting a clinical study on Allium cepa and in silico data on a possible mode of action.

While this is an interesting study some issue remain to be clarified and must be addressed.

The authors report a clinical study. To improve the value of the manuscript the authors should stringently follow the CONSORT criteria/items. Please recheck.

The CONSORT chart which is currently presented as a supplement should be included in the main document.

The authors have to report  how the number to treat was calculated.  

How was randomization undertaken.

No information is given whether patients, applicant and observer were blinded to treatment.

The mode of application is not described (only that it was 3 month).

The concentrations of Allium cepa and HA in the device have to be given.

How was adherence to the protocol and application supervised.

I have great difficulties with the presented statistics. All scar scales are ordinal scales . Therefore  calculation of a mean is incorrect. Please correct and also subsequent statistical tests.   

The discussion is incomplete. The authors have to discuss the finding that no significant difference was seen at 12 weeks. Were patients not following the protocol? ( see my point on protocol adherence).

Furthermore in the discussion a section on shortcomings is completely missing.

This should discuss the results at 12 weeks.

It should discuss that scar maturation is not complete at 12 weeks.  

It should discuss the choice of controls. This said the  sentence  “ Our clinical insight demonstrated the efficacy of quercetin of Allium cepa medical device” is not justified. The study design does not allow this statement as the appropriate control would have  been a class I pullulan based medical device and HA gel without Allium cepa.

Similar, the statement that “Quercetin inhibition of fibroblasts proliferation and contraction sustain the favorable results obtained by using medical devices containing Allium cepa and HA to prevent hypertrophic scar formation in post-surgical wounds"is not justified  and simply not correct.

The word “pivotal” should be removed throughout the manuscript.

The result at 12 weeks should also be given in the abstract ( it does not minimize the study).

Author Response

First of all,

we would like to thank both reviewers for their constructive comments. Below are our answers, item by item.

With our best regards

  • The authors present and interesting paper reporting a clinical study on Allium cepa and in silico data on a possible mode of action. While this is an interesting study some issue remain to be clarified and must be addressed.

Dear Reviewer, thank you for your interest in our manuscript and for the profitable comments to improve its quality.

  • The authors report a clinical study. To improve the value of the manuscript the authors should stringently follow the CONSORT criteria/items. Please recheck.

Dear reviewer, thank you for your suggestion. We have modified the manuscript according to your suggestions and to CONSORT criteria.

  • The CONSORT chart which is currently presented as a supplement should be included in the main document.

Dear reviewer, thank you for your suggestion. We have reported the CONSORT chart in the main document.

  • The authors have to report how the number to treat was calculated.  

Dear reviewer, thank you for your suggestion. The sample size was calculated using two independent means which were acquired from previous research. We have added this information in the material and methos section with references.

The sample size was calculated using two independent means which were acquired from previous research [14,15].

  • How was randomization undertaken.

Dear reviewer, thank you for your suggestion. We specify, according to CONSORT criteria and your suggestion about randomization in the method section.

“The randomization sequence was created using Excel version 2306, with a 1:1 allocation of the subject in one of the two arms by sub-investigator G.C. . Sub-investigator T.C. enrolled the participants in the Dermatological clinic of the Tor Vergata University Hospital. Sub-investigator R.G. assigned participants to interventions using simple randomization in Excel version 2306. Medical device samples, previously numbered by sub-investigator G.C., were given to subjects by sub-investigator T.C.”

  • No information is given whether patients, applicant and observer were blinded to treatment.

Dear reviewer, thank you for your suggestion. Since it was not a blind study, the patients, applicant and observer were aware of the treatment performed. We reported the lack of blinding as a limitation of the study in the dedicated section.

  • The mode of application is not described (only that it was 3 month).

Dear reviewer, thank you for your suggestion. We reported and clarified that all patients involved in the study applied topical medical devices for 3 months on post-surgical area.

  • The concentrations of Allium cepa and HA in the device have to be given.

Dear reviewer, thank you for your suggestion. We have added this information in the abstract and in the materials and methods section.

  • How was adherence to the protocol and application supervised.

Dear reviewer, thank you for your suggestion. To improve our manuscript, we have added the following sentences in the material and methods section.

“The application of the products was recorded by sub-investigator T.C. and R.G by means of a paper diary provided to the subjects in which they reported the application and/or non-application of the medical devices. Furthermore, during the enrolment visit sub-investigator T.C. explained how to apply the medical device (after cleansing the area, apply a layer of medical device with circular movements until completely absorbed).”

  • I have great difficulties with the presented statistics. All scar scales are ordinal scales. Therefore, calculation of a mean is incorrect. Please correct and also subsequent statistical tests.   

Dear reviewer, thank you for your suggestion. We have modified the statistical section and related tests. We previously used a parametric test based on the central limit theorem. Following your advice, we analysed data using a non-parametric test for ordinary variables. The results are comparable to those obtained using the parametric test.

The statistical paragraph in the material and method section has been modified as follows.

The statistical analysis of the data was carried out by applying parametric or non-parametric tests depending on the distribution of the data that was going to be obtained. Results were reported as average ± SE, median values or percentages, considering parameter type. Non-parametric Krustal-Wallis test was used to evaluate the variation of clinical parameters in the pre-established observation times (T0, T1, T2 and T3) for each treatment, whereas comparison between treatments was performed via Mann-Whitney analysis and chi-squared test (χ2 test). The study was conducted on a group of subjects selected through specific inclusion and exclusion criteria according to the scientific rationale of the protocol. SPSS v.27 software was used for statistical analyses. The differences were considered statistically significant for p-values <0.05.”

  • The discussion is incomplete. The authors have to discuss the finding that no significant difference was seen at 12 weeks. Were patients not following the protocol? (see my point on protocol adherence).

Dear reviewer, thank you for your suggestion. We added the results at T3 (12 weeks of treatment) in the results and discussion section. We have corrected line 102 that could create confusion to the reader. Moreover, all patients followed the protocol thanks to simplicity of application of the products in the studio and to the post-surgery aesthetic interest. We have added this sentence in the results section.

Moreover, all patients followed the protocol thanks to the simplicity of application of the products in the studio and to the post-surgery aesthetic interest”.

  • Furthermore in the discussion a section on shortcomings is completely missing. This should discuss the results at 12 weeks. It should discuss that scar maturation is not complete at 12 weeks.  

Dear reviewer, thank you for your suggestion. We have stressed the study limitation at the end of the discussion. We added few lines about scar maturation as limitation of the study at the end of the discussion.

  • It should discuss the choice of controls. This said the sentence “Our clinical insight demonstrated the efficacy of quercetin of Allium cepa medical device” is not justified. The study design does not allow this statement as the appropriate control would have been a class I pullulan based medical device and HA gel without Allium cepa.

Dear reviewer, thank you for your suggestion. We have modified the sentence in order to follow the aims of the study as reported in the material and method section and eliminated the aforementioned sentence.

  • Similar, the statement that “Quercetin inhibition of fibroblasts proliferation and contraction sustain the favorable results obtained by using medical devices containing Allium cepa and HA to prevent hypertrophic scar formation in post-surgical wounds" is not justified and simply not correct.

Dear reviewer, thank you for your suggestion. We have eliminated the sentence in the conclusion section and improved the hypothesis that quercetin could prevent hypertrophic scar in the discussion section.

The use of plant extracts in wound healing is associated with a satisfying outcome in most cases. In this field, flavonoids are generally abundant in natural extracts and the high content of quercetin is an interesting feature. In fact, quercetin has been studied in various wound healing conditions, alone or combined with other phytochemicals, proving to be a valid option in the management of wound repair [37]. Allium cepa presents different active molecules including gallic acid, quercetin, protocatechuic acid and kaempferol [38]. Among them, quercetin, is the major flavonoid component with concentrations ranking until 5110 µg/g, more than 40 folds the concentration of other isolated compounds such as gallic acid and kaempferol. Moreover, previous publications investigated the role of quercetin in wound healing and scar prevention. Karakaya et al. [39] evaluated the ethnopharmacological use of Epilobium angustifolium, E. stevenii and E. hirsutum by using in vivo and in vitro experimental models, and to identify the active wound-healer compound (s) and to explain the probable mechanism of the wound-healing activity. Their in vitro tests showed that quercetin and its glucosides could be the compound responsible for the wound-healing activity due to their significant anti-hyaluronidase, anti-collagenase, and antioxidant activities. In parallel, Özbilgin et al. [40] evaluated E. characias subsp. wulfenii extract in wound healing. Quercetin et its glucoside derivates were found to be the highest compound in the extract of E. characias subsp. wulfenii, which might be the active principles in anti-inflammatory and wound-healing activities. Mi et al. [41] evaluated the effect of quercetin on wound healing by analyzing wound healing rate, pro-inflammatory cytokines in mice. Their observations come from the use of Oxytropis falcata, a wild-growing plant mainly distributed in an altitude of 2700–4300 m of China, used in treating inflammation, injury and bleeding. They demonstrated that quercetin can promote the proliferation and migration of fibroblasts and HaCat cellsl. Furthermore, quercetin has been shown to promote wound healing in mice by inhibiting the inflammatory response and increasing the expression of the growth factor, as well as Wnt/β-catenin pathway and telomerase RNA catalytic elements (TERT).

  • The word “pivotal” should be removed throughout the manuscript.

Dear reviewer, thank you for your suggestion. According to your observation, we have modified the main document and removed the word “pivotal”.

  • The result at 12 weeks should also be given in the abstract (it does not minimize the study).

Dear reviewer, thank you for your suggestion. We have added the results at 12 weeks in the abstract.

Round 2

Reviewer 2 Report

The manuscript  has been improved; I have no further queries.